# Smart Water Grid Research Group Project: An Introduction to the Smart Water Grid Living-Lab Demonstrative Operation in YeongJong Island, Korea

**Kang-Min Koo [1], Kuk-Heon Han [2], Kyung-Soo Jun [1], Gyumin Lee [3] and Kyung-Taek Yum [2,*]**

[1] Graduate School of Water Resources, Sungkyunkwan University, Suwon 16419, Korea; koo00v@skku.edu (K.-M.K.); ksjun@skku.edu (K.-S.J.)
[2] Smart Water Grid Research Group, Sungkyunkwan University, Suwon 16419, Korea; kuk0904@daum.net
[3] Construction and Environmental Research Center, Sungkyunkwan University, Suwon 16419, Korea; leegyumin@gmail.com
[*] Correspondence: kwfyum@gmail.com; Tel.: +82-31-290-7645

**Abstract:** In South Korea, in line with the increasing need for a reliable water supply following the continuous increase in water demand, the Smart Water Grid Research Group (SWGRG) was officially launched in 2012. With the vision of providing water welfare at a national level, SWGRG incorporated Information and Communications Technology in its water resource management, and built a living lab for the demonstrative operation of the Smart Water Grid (SWG). The living lab was built in Block 112 of YeongJong Island, Incheon, South Korea (area of 17.4 km², population of 8000), where Incheon International Airport, a hub for Northeast Asia, is located. In this location, water is supplied through a single submarine pipeline, making the location optimal for responses to water crises and the construction of a water supply system during emergencies. From 2017 to 2019, ultrasonic wave type smart water meters and IEEE 802.15.4 Advanced Metering Infrastructure (AMI) networks were installed at 527 sites of 958 consumer areas in the living lab. Therefore, this study introduces the development of SWG core element technologies (Intelligent water source management and distribution system, Smart water distribution network planning/control/operation strategy establishment, AMI network and device development, Integrated management of bi-directional smart water information), and operation solutions (Smart water statistics information, Real-time demand-supply analysis, Decision support system, Real-time hydraulic pipeline network analysis, Smart DB management, and Water information mobile application) through a field operation and testing in the living lab.

**Keywords:** SWG; SWGRG; SWM; WDN; living lab; YeongJong Island

## 1. Introduction

The ultimate goal of Water Distribution Network (WDN) management is to provide consumers with a safe, reliable, and sustainable water supply. However, the existing WDNs face problems such as supply shortages, water quality degradation, increased energy consumption, climate change, and aging facilities. A modern water supply system is required to reduce the current water problem; however, upgrading the existing WDNs is costly and time consuming. Therefore, improving the existing WDNs with smart components such as sensors, networks and an integrated operations center allows water utility companies to monitor and control the water supply in real time, which is a more cost-effective and sustainable approach [1]. A Smart Water Grid (SWG) is capable of reducing water problems without compromising water sustainability, enhancing the efficiency of WDNs through the integration of Information and Communications Technology (ICT) and a conventional water management system. In other words, with an

SWG, the incorporation of ICT enables real-time monitoring, the measurement and modeling of water consumption, and reliable water management information, which can be reflected in the related decision making [2]. In particular, an SWG resolves the problem of uncertainties in the existing human inspection method and facilitates a demand analysis through a real-time trend of water consumption, thereby supporting the understanding of water consumption characteristics of consumers [3]. In addition, in the event of an incident (e.g., leakage or a malfunctioning meter) in the water pipeline network, an SWG improves management efficiency, reduces maintenance costs through fast responses and enables an effective pressure management of the water pipeline network. In addition, water consumption data collected in real time can be used to encourage water savings for consumers, and the deviation between real water consumption and billed usage can be improved [4].

Despite these advantages, the reason for the low penetration rate of SWGs has been the high initial installation cost, but they are currently regaining their momentum owing to the rapid development of the low-cost, high-efficiency Internet of Things (IoT) technology [5]. Accordingly, research on the application of an SWG has been underway worldwide. In particular, Singapore, Australia, the European Union (EU), the United States (US), and South Korea have taken the lead in implementing smart water management by incorporating SWG technology.

Singapore is a representative example of a country with a water shortage, importing 40% of water from Malaysia. Therefore, the country developed the SWG Roadmap, led by the government, to secure stable water resources and water supply, and international joint research has been carried out. In particular, since 2013, 346 sensor stations have been built and operated in Singapore by linking the WaterWiSe platform with the SCADA system of the Public Utilities Board (PUB) [6]. WaterWiSe conducts real-time monitoring of the water quality, such as the pH, conductivity, and turbidity of the water supply and pressure to support decision-making in the management and operation of a WDN. As a result of the test-bed operation, water pressure management has been achieved through a post-event analysis of the water main.

Australia has been implementing the South East Queensland (SEQ) water grid project since 2008 to tackle the problems of extreme drought. A bulk water supply network connecting all of the 12 dams, 36 water treatment plants, three purified recycled water treatment plants, one desalination plant, 28 reservoirs, 22 pumping stations, and 600 km of pipelines were constructed to ensure the stability of the water supply [7].

The EU carried out its ICT Solutions for Efficient Water Management (ICeWater) project and the Water Innovation through Dissemination Exploitation of Smart Technologies (WIDEST) project supported by the European Commission. These projects are aimed at improving the energy efficiency of urban WDNs through real-time water consumption monitoring and reducing water loss by developing leak detection technologies for pipelines [8]. In particular, testbeds were built in Milano, Italy and in Timisoara, Romania, and research applying AMI network was conducted.

To supply freshwater to the western states of the US, including the Colorado River Basin and areas with water shortages from drought, a water supply plan using large rivers in the Midwest, such as the Mississippi River, which experiences severe flood damage, was established through the National Smart Water Grid (NSWG) project [9].

In South Korea, for the implementation of SWG technology, the country is taking pioneering steps to build a hyper-connected smart city by carrying out a local water supply modernization project and constructing and operating a water industry cluster. To this end, the Smart Water Grid Research Group (SWGRG) was launched in 2012 to build an intelligent water management system using alternative water resources with the aim of reducing water and energy consumption. With the convergence of ICT and water resource management technologies, the core technologies for an SWG with intelligent microgrids were developed and field operation and tests were conducted in a SWG living lab [10].

As above, SWG is being actively conducted worldwide. However, there are few cases of developing detailed element technologies and operation solutions through the operation of living labs. Therefore, this study introduces the development of detailed element technologies for SWG management and an operation solution for South Korea through a field operation and tests in a living lab. We would also like to show examples of what issues there are in living lab operations.

## 2. SWGRG Project Overview in Korea

### 2.1. Objectives

With the aim of accomplishing water welfare at a national level, SWGRG incorporates advanced ICT to develop and construct a highly efficient next-generation water management infrastructure and system and verify the newly developed system through a field operation. To this end, an SWG architecture (Figure 1) is developed as described below, which is an integrated intelligent water management system that incorporates the ICT from water-intake to the end-user. With an SWG, we aim to address water problems through multi-dimensional approaches such as ensuring a reliable supply of water resources in response to climate change and urbanization, resolving temporal and spatial gaps, a stable supply of necessary water resources with adequate water quality, the design and management of low-energy and high-efficiency water resource facilities, and active responses to the paradigm shifts in the global water industry [11].

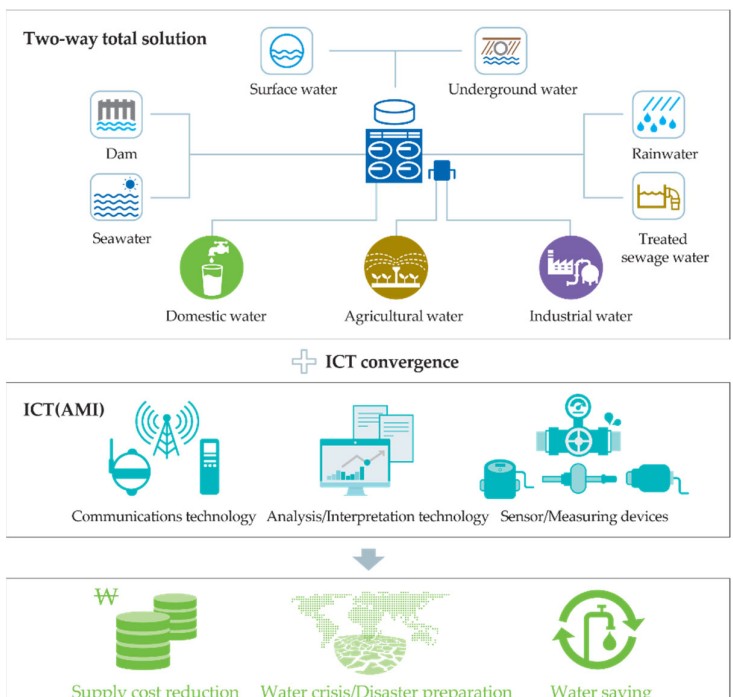

**Figure 1.** Representative diagram of SWG architecture.

### 2.2. Project Period

In accord with the SWG architecture described above, Phase 1, a "Water grid intellectualization" project, was carried out for 56 months from July 2012 to February 2017 for the development of the technological elements of a high-efficiency next-generation water management infrastructure system incorporating advanced ICT technologies. Thereafter, Phase 2, "An empirical study on the advancement of an SWG facility O&M project" was conducted for 33 months from April 2017 to December 2019, and using the

element technologies developed in Phase 1 water management technology was further advanced through the field operation of a living lab (Table 1).

**Table 1.** SWGRG project period and budget by phase.

| Phase | Period | Budget |
|-------|--------|--------|
| 1 | July 2012–February 2017 (56 months) | 31.2 billion KRW (Govt. contribution: 23.0 billion KRW) |
| 2 | April 2017–December 2019 (33 months) | 3.47 billion KRW (Govt. contribution: 2.53 billion KRW) |

### 2.3. Project Team and Task

For the SWGRG project, 40 research institutes including Incheon National University participated in phase 1 to carry out three main tasks (secure water resources/distribution, construction and management of an SWG integration system, and ICT-based bi-directional optimal management), and 13 institutes including Sungkyunkwan University participated in Phase 2 to carry out four main tasks (development of a multi-source water smart operation platform and application, AMI-based smart water management, advancement of O&M technologies, and overseas marketing strategy) (Figure 2).

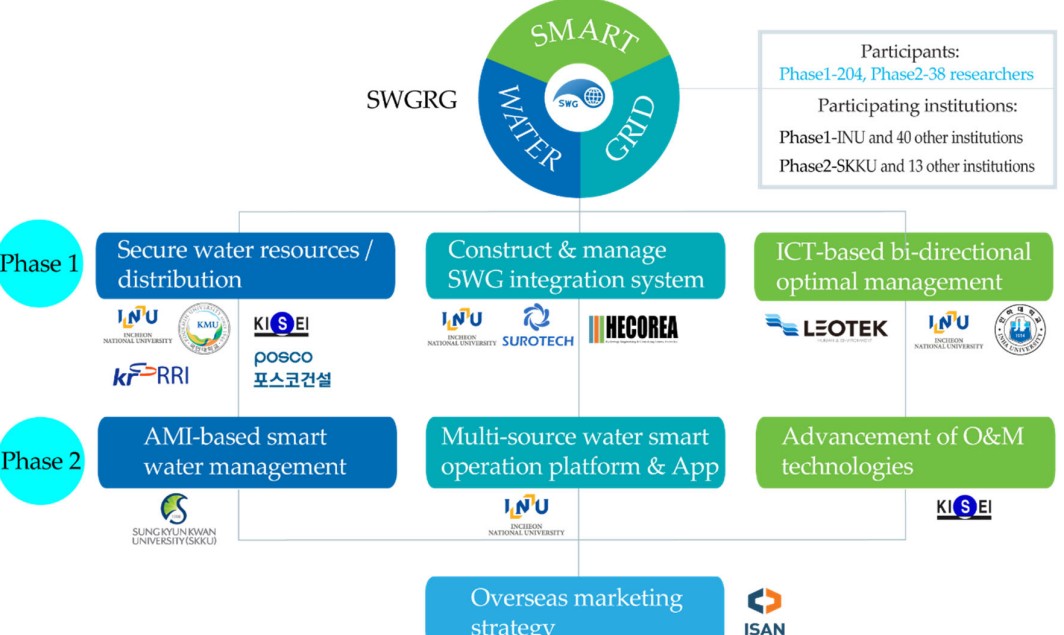

**Figure 2.** SWGRG project team and task.

### 2.4. Living Lab for SWG

YeongJong Island, located in Jung-gu, Incheon, South Korea, was selected as a living lab construction site for the field operation of an SWG. Incheon International Airport, a hub for Northeast Asia, is located in YeongJong Island (Figure 3). In addition, because water is supplied to the distributing reservoir from the Gongchon water filtration plant to YeongJong Island through a single submarine pipeline to the distributing reservoir, the location is optimal for the application of ICT technologies in response to water crises and the construction of a water supply system during an emergency. YeongJong Island is located approximately 30-km west of Seoul, and as of 2018 the population is ~75,000, and the total area is ~104 km$^2$. Block 112 of YeongJong Island has a population of ~17,000 and an area of ~17.4 km$^2$ and is composed of a new airport city (Unseo-dong) and an

administrative district, Unbuk-dong. The WDN of YeongJong Island is managed by the Incheon waterworks headquarters, and there is one distributing reservoir in Block 112. In addition, the total pipeline length is approximately 68 km, and the total number of water meters is 958, with a daily water supply of approximately 8000 m$^3$ and a revenue water ratio of 73.2%.

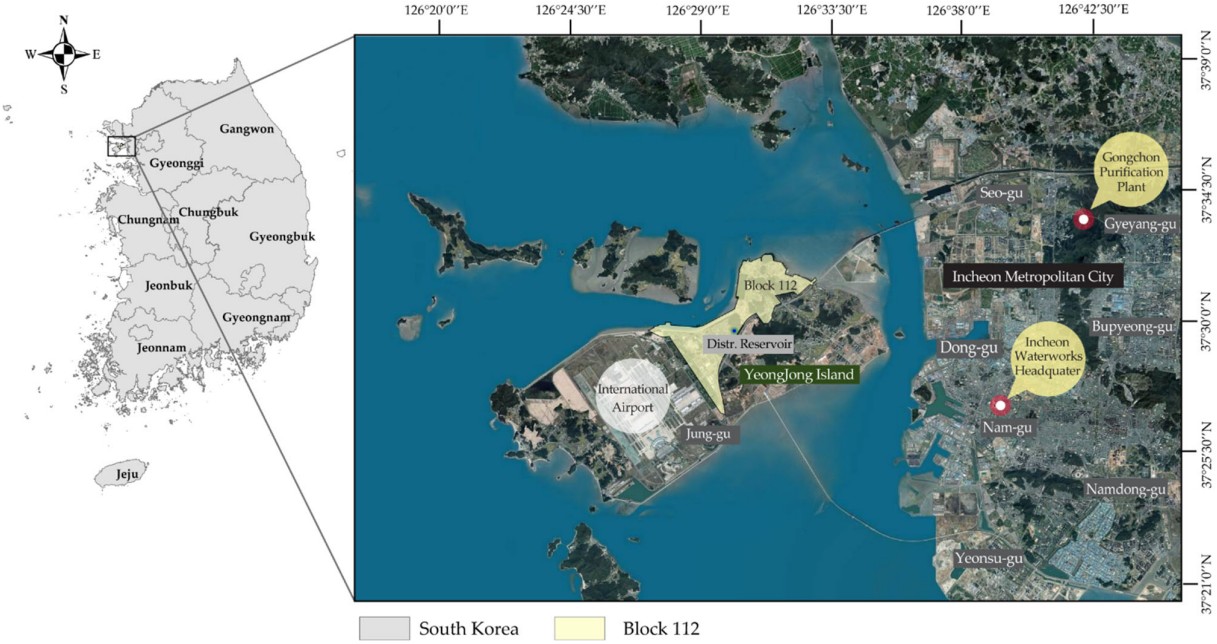

**Figure 3.** Location of YeongJong Island and SWG WDN block 112.

## 3. Operation Technique

The core element technologies developed in Phase 1 are an intelligent water source management and distribution system, smart WDN planning/control/operation strategy establishment, AMI network and device development, and integrated management of bi-directional smart water information.

### 3.1. Intelligent Water Source Management and Distribution System

With SWGRG, a real-time intelligent water source management and supply system was developed. To this end, various meters such as a multi-point flow meter and image-based water level gauge were installed for collection and adjustment of the water resource data, and a demand-based supply capacity for the available quantity was evaluated based on the water source. In addition, water treatment process technologies were developed in which, using real-time measurement data of five water quality items (pH, water temperature, alkalinity, electrical conductivity, and turbidity) and a multivariate statistical analysis technique, abnormalities in the water quality were assessed to determine the acceptance to the selective water-intake. By monitoring the pump status, a selective water-intake is possible through the optimal combination of the number of pumps depending on the pump head and the failure status of the pumps. At this time, water supply processes for regular days and emergencies were established, and depending on the state of the water source, through a calculation of the selective water-intake cost, the economic effects, i.e., an improvement in the water resource independence by 5%, and 5% energy cost saving for the supply and distribution pumps, were achieved (see Figure 4 [10]).

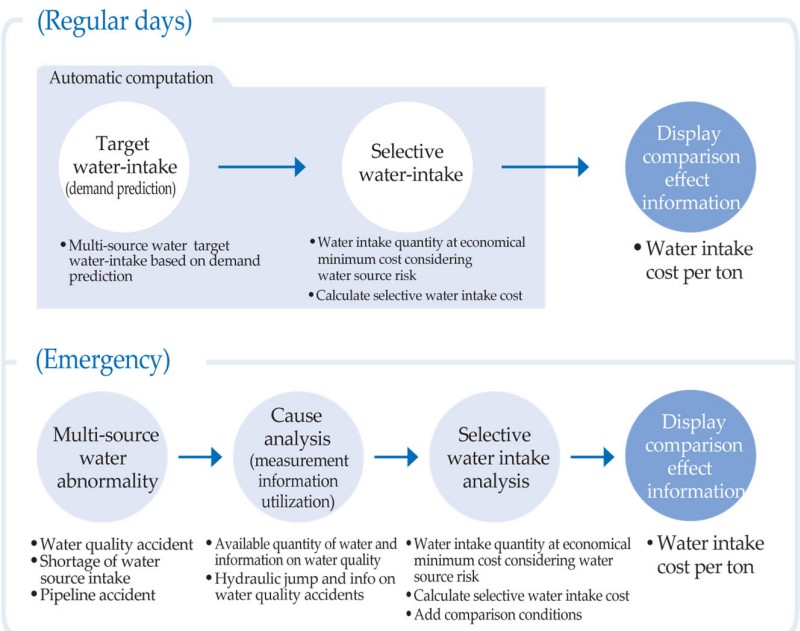

**Figure 4.** Multi-source water selective water intake process for regular and emergency water supply.

## 3.2. Establishment of Smart WDN Planning/Control/Operation Strategy

With real-time measurements through an SWM based on Advanced Metering Infrastructure (AMI), multi-source water was linked with a water-loop for data acquisition in real-time, establishing a water supply plan capable of managing water withdrawal, distributing the water supply, pump scheduling, and leak management (Figure 5). Water is supplied from source through a water intake plant and a water filtration plant, and to a block distributing reservoir. At this time, by forecasting the end-user demand, the optimal water level of the distributing reservoir is determined, and water-intake management is possible through multi-source combinations according to the production volume reflecting the demand, thereby reducing the power and chemical treatment costs. With the AMI device, demand forecast by use is possible, and using the end-use water consumption data, adequate water pressure management is possible through a hydraulic water pipeline network analysis.

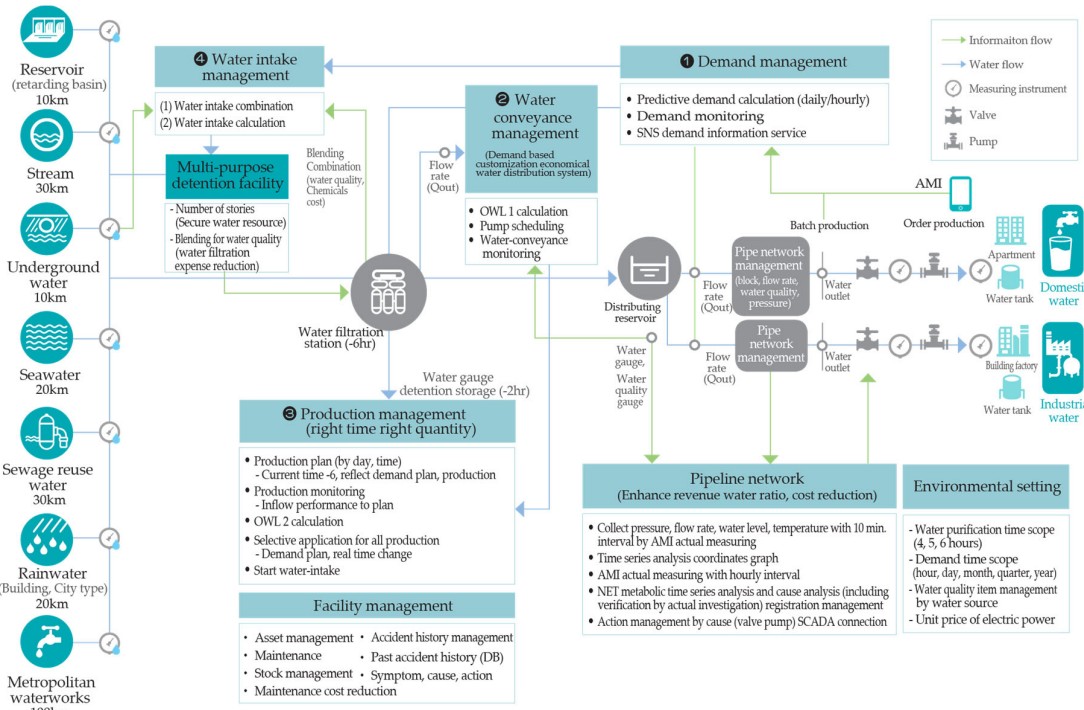

**Figure 5.** Establishment of the strategic plan for smart water supply network planning/control/operation.

### 3.3. AMI Network and Device DevelopmentDevelopment

The development of AMI components and networks for data collection, which forms the basis of the water information service, is defined as an AMI network that is bi-directional and accommodates end-users with high speed and with a large capacity [12,13]. The AMI network is composed of an End Device (ED) or an SWM that collects water consumption data, an Outdoor Home Display (OHD) that is installed such that the data can be checked from the outside, an End Device Manager (EDM) that manages each Smart Meter Device (SMD) and OHD, and a Network Coordinator (NC) that transmits the measured data to the server through a mobile communication network [12].

The SWM provides various information on the water consumption of end-users in a bi-directional manner. The SWGRG has developed an SWM in which even sensors are operated electronically by adopting a differential Time-of-Flight (dToF) method using an ultrasonic wave type (Figure 6). The advantages of SWM development include price competitiveness compared to existing prototypes, a reduction in power costs, a 1.5–2-fold improved measurement precision, and a structure resistant to freezing and bursting [10].

For existing wireless inspection systems, a low-speed based remote inspection system for use in the 424-MHz band and a wireless remote inspection with 2.4-GHz Zigbee are typically used. However, it is difficult to overcome environmental restrictions such as various types of interference owing to the nature of these frequencies, and because it is dependent on unidirectional communication from the SMD to the NC, it is impossible to expand this system to various inspection services. In addition, fast detection and responses to errors and failure situations are difficult to achieve with this system. Therefore, SWGRG developed a multi-channel cluster AMI network by applying IEEE 802.15.4 Smart Utility Network (SUN) technology [12].

For field operation of the living lab, AMI devices such as five pressure gauges (three in Unbuk-dong and two in Unseo-dong), 527 SWMs (approximately 55% of the total number of water meters), one multi-purpose water quality gauge and three flowmeters were installed (Figures 7 and 8, [10]). Out of 958 consumer sites in all of Block 112, 527 have an SWM installed, and there are eight types of pipeline diameter ranging from 15 to

100 mm. According to the billing standards, these are classified into 387 for domestic use, 138 for general use, and two for water tanks. Among them, there were 279 domestic SWMs of 15 mm in diameter, accounting for approximately 52.9%. The water consumption data collected by the SWM is sent on hourly basis to the SWG integrated operation center server built in the waterworks headquarters of Incheon Metropolitan City.

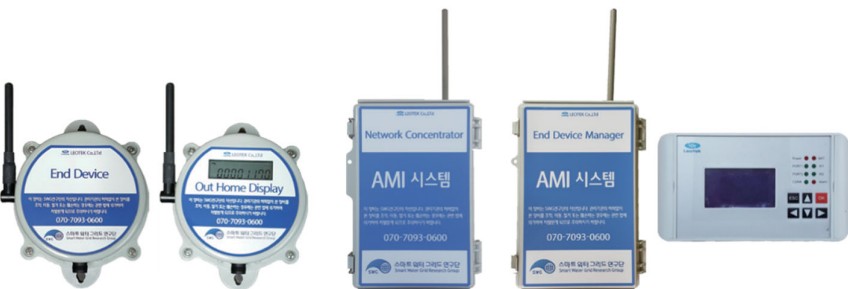

**Figure 6.** Developed AMI network device (SWM, OHD, NC, EDM).

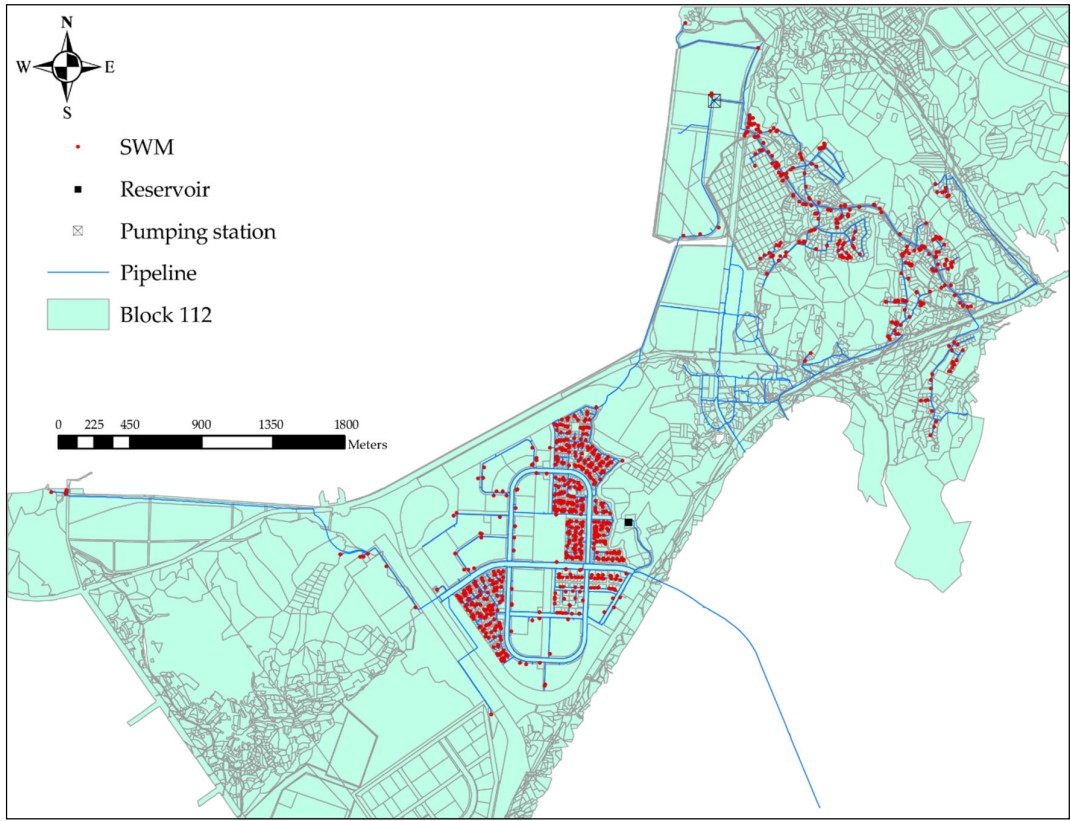

**Figure 7.** Deployment of smart water meter in Block 112.

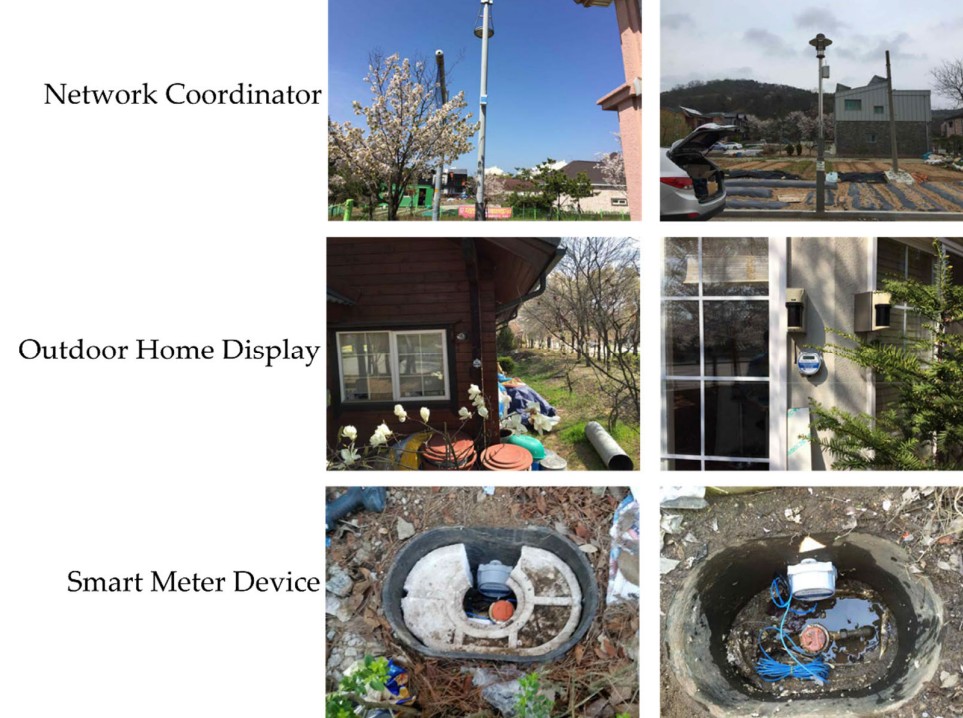

**Figure 8.** Setup of AMI network device.

### 3.4. Integrated Management of Bi-Directional Smart Water Information

For an integrated and efficient operation and management of water resource data, meteorological data, real-time monitoring data, and program result data, which are currently managed and operated by separate organizations and systems, the design and construction of a systematic DataBase (DB) are imperative [14]. Therefore, in conjunction with an SWG DB, a DB of national and related organizations (Korea Metrological Administration (KMA), Incheon International Airport, Incheon Metropolitan City), and real-time AMI measurement data, a smart integrated DB was constructed for stable and efficient linkage and operation of input/output data of the program used in the SWG using open API, file to DB, and DB to DB methods (Figure 9). A separate closed SWG network was built inside the Incheon waterworks headquarters for construction of an integrated SWG DB, and the SWG DB server used a separate communication gate for connection with the Incheon water management DB server. At this time, a collection server is operated by configuring a demilitarized zone to store data from external related organizations and the measurement data. The meteorological data and forecast data of the KMA, one of the related organizations, are connected to the external data collection server through the open API, and the data on the distributing reservoir water level and reclaimed water use of the Incheon International Airport Co. are collected and directly sent to the server. In addition, the AMI network data installed in Block 112 of YeongJong Island, and multi-source hydraulic monitoring data (flow rate, water level, and water quality) in YeongJong Island, are also collected by the server. Data built on the external collection server are connected to the SWG integrated DB of the SWG closed network through a firewall using the enterprise security line for real-time connection and operation [14].

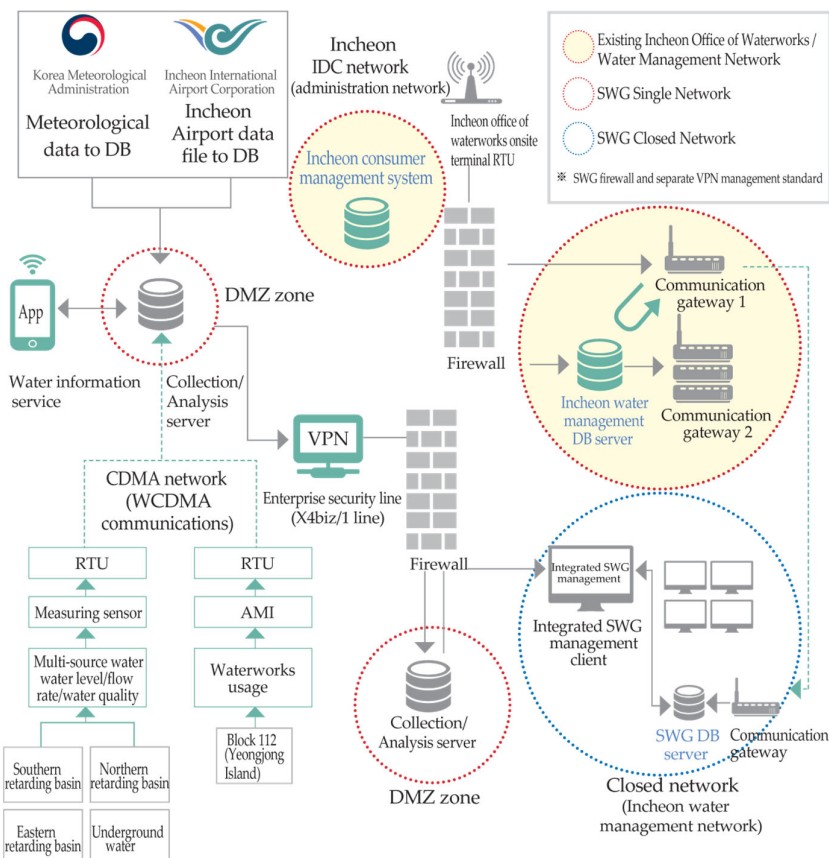

**Figure 9.** Block diagram for integrated management of bi-directional smart water information.

## 4. Operation Solution

SWGRG has developed an operation process from AMI data to integrated operation solution development as a preliminary task for developing the field operation solution of a living lab (Figure 10). The statistical analysis of the AMI data collected in real time provides water consumption data through an analysis of the cause of outliers in the water consumption data, receiving rate, pre-processing, usage, pipe diameter, day of the week, and region. With real-time water consumption data as input, statistical methods and Machine Learning (ML) techniques are applied to forecast future water demand. A real-time water pipeline network analysis enables proper water pressure management using Geo-graphical Information Systems (GIS) and the EPANET engine (United States Environmental Protection Agency), and an optimal operation of distributing reservoir is possible by identifying households where water supply has been stopped in case of an incident such as a leak. For decision making support, the optimal water supply for each water source must be determined according to calculations of the available quantity based on demand. Water supply management when considering the available amount of water calculated by applying GIS, and pump scheduling optimization through an analysis of the distributing reservoir supply, can provide a support in the decision making. With the design of the SWG integrated operation solution UI and connection with real-time measurement data, feedback is received through actual operation, and the problems are corrected or improved accordingly.

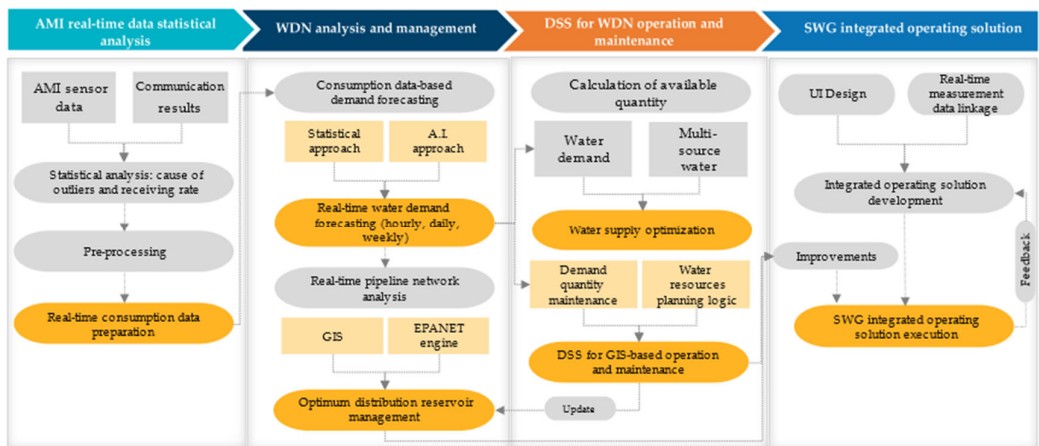

**Figure 10.** Procedure for developing SWG integrated operation solution.

The result finally obtained from the field operation of the SWG living lab is the SWG integrated operation solution, called a Smart Water Management Integrated Solution. The solution is composed of six main functions that link the AMI sensor data and the reference layer (GIS, HMI, real-time hydraulic modeling, smart DB). The six functions for achieving proper water management are the smart water statistics information (SWG-STAT), real-time demand-supply analysis (SWG-DSM), decision support system (SWG-DSS), real-time hydraulic pipeline network analysis (SWG-HyNet), smart DB management (SWG-DBM), and water information mobile application (SWG-App.) (Figure 11).

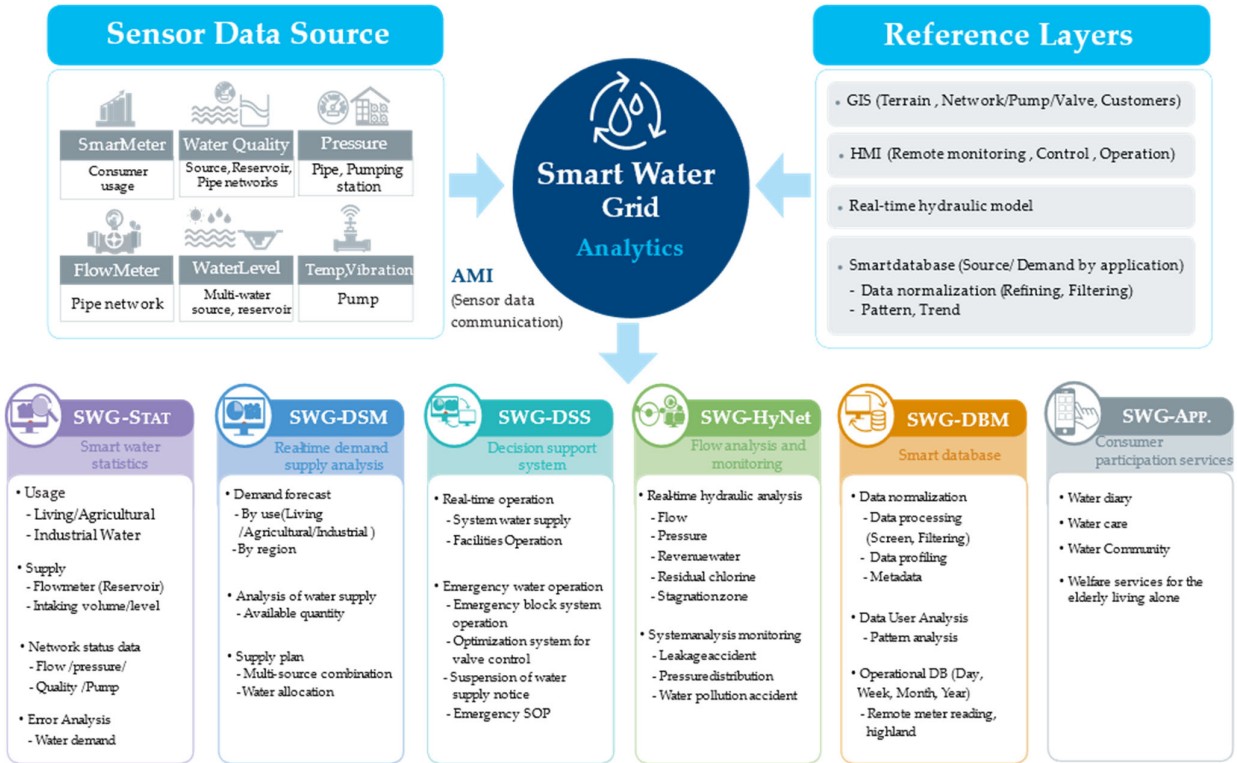

**Figure 11.** Design and construction of smart water management integrated solution based on real-time water information.

### 4.1. SWG-STAT

With smart water statistics information, real-time measurement data from AMI sensors installed at a water intake plant, a water filtration plant, a distributing reservoir, and at the locations of the end-users can be retrieved and viewed with ease (Figure 12). In addition, hourly, daily, monthly, and a specified period of information on water consumption and water supply by use (domestic, agricultural, and industrial), the status of measuring instruments, the energy consumed, and the revenue water ratio are provided. In particular, as shown in Figure 12, it is possible to check in real-time water withdrawal based on the source (stream, underground water, retarding basin, reservoir, sewage reuse water, and seawater) and water level, and the real water consumption based on use and average consumption. AMI data include outliers (missing data, incorrect measurement data) according to sensor abnormalities or communication errors, and these missing and erroneous data are corrected by applying a pre-processing technique [15] of big data combining a lagged k-nearest neighbor (k-NN) with fast Fourier transform.

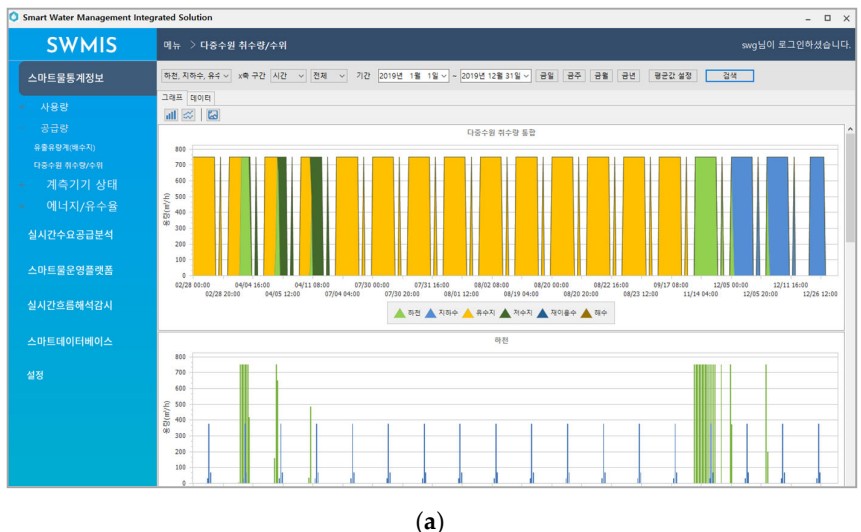

(**a**)

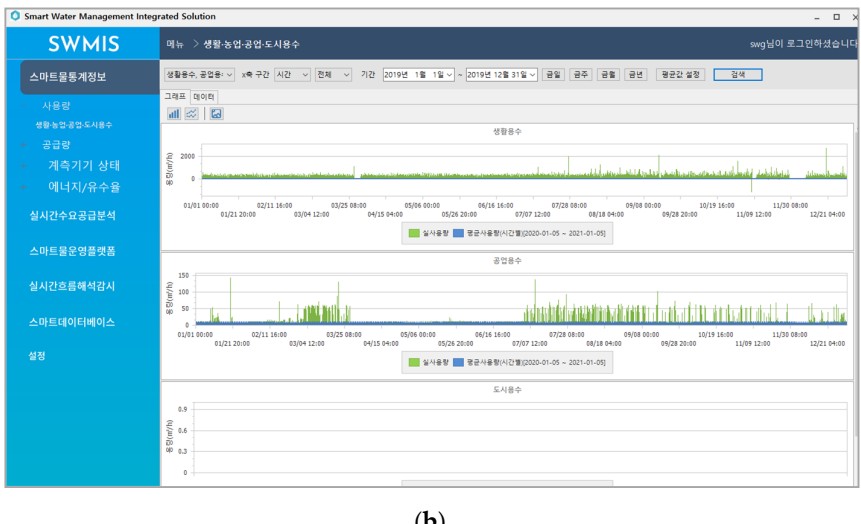

(**b**)

**Figure 12.** Smart water management integrated solution dashboard: SWG-STAT. (**a**) Multi-source water selective withdrawal monitoring: i.e., integration of multiple-source water intake and intake from the stream. (**b**) Multi-source water selection based on real-time demand: domestic, industrial, and urban water.

### 4.2. SWG-DSM

In a real-time supply–demand analysis, a water distribution and supply plan can be established through the analysis of a multi-source water supply according to the forecasted water demand by use and region. The water demand of end-users can be forecast in real time by combining the statistical seasonal autoregressive integrated moving average method [16] and machine learning k-NN [17]. The results are provided on a daily, weekly, and monthly basis and a comparative analysis between the forecasted water demand and real water demand is possible. As shown in Figure 13, the results of the demand forecast consist of demand at end-use, the distributing reservoir demand, and demand by use. Using these results, the available quantity, water withdrawal, and real water supply of multiple sources can be analyzed. The combination of selective water withdrawal of multiple sources uses the cost function for calculating the lowest supply cost, and the optimal solution is found using the Harmony search optimizer [18].

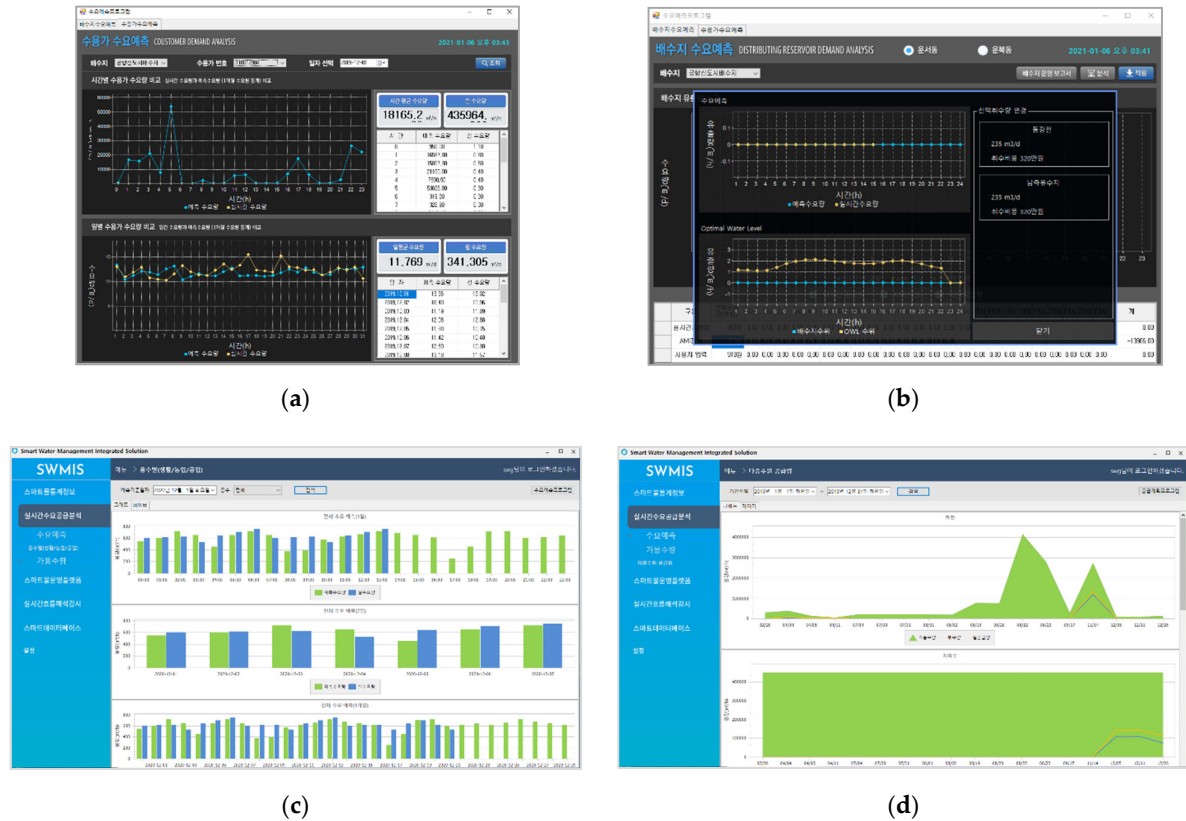

**Figure 13.** Smart water management integrated solution dashboard: SWG-DSM. (**a**) Comparison between forecast and real water demand at end-user. (**b**) Comparison between forecast and real water demand at distribution reservoir. (**c**) Demand forecast by use: hourly, weekly and daily total forecast and actual demand. (**d**) Multi-source water supply plan: stream, underground.

### 4.3. SWG-DSS

With the decision support system, the WDN operator monitors the operation status on regular days for integrated water management, and in the event of an emergency, such as a water supply shut-off, multi-source pollutant inflow, or drought, the system supports prompt decision making on the water distribution, supply, and shut-off [19]. On regular days, the available number of multiple sources, water inflow, distributing reservoir water level, and BOD water quality are monitored (Figure 14), and when an emergency situation

arises, the operator can respond according to the scenario by displaying the emergency screen, and the consumer can be informed of the emergency through an SNS alert.

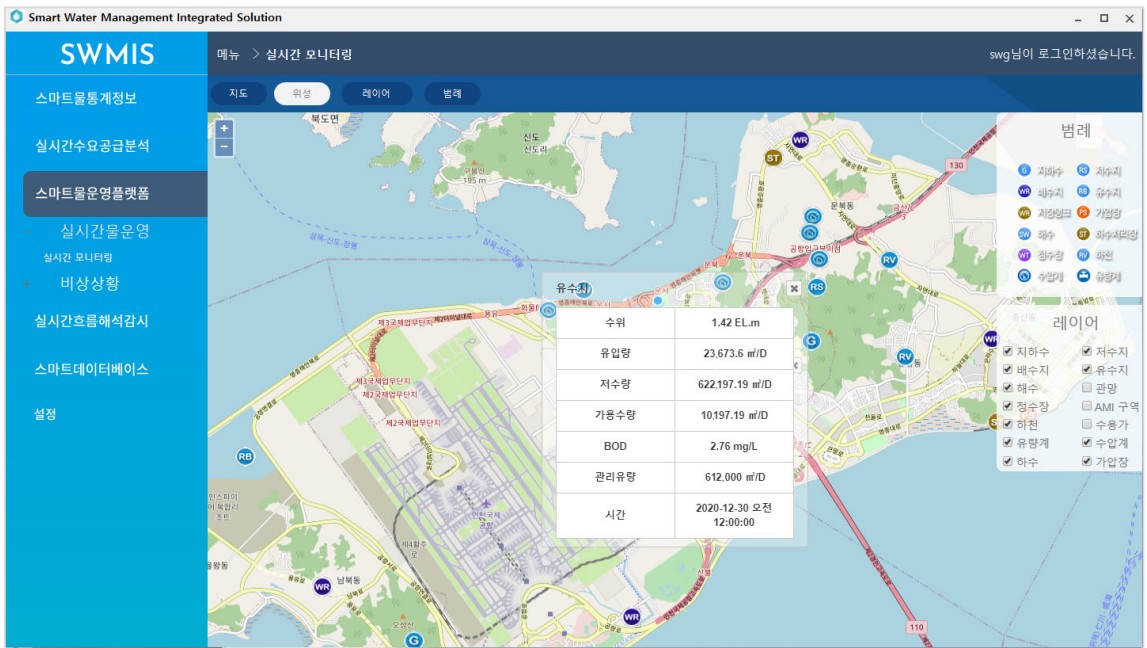

**Figure 14.** Smart water management integrated solution dashboard: Real-time WDN operation by SWG-DSS (water level, inflow, outflow, storage volume, available quantity, BOD, management flow, and time).

### 4.4. SWG-HyNet

A real-time hydraulic pipeline network analysis consists of an interface visualized through GIS and hydraulic modeling using the EPANET engine. The water pipeline network is composed of nodes (including end-users), pipes, distributing reservoirs, valves, and pumps. The EPANET engine can conduct a simulation of the flow rate of each node and pipe, water pressure, water quality behavior, and residence time [20]. As shown in Figure 15, when demand-driven modeling is performed with the forecast water demand of each end-use in SWG-DSM as input data, the hydraulic pressure distribution can be determined for the points of interest in the block. Identifying the leak location is difficult with the existing WDNs, but SWG-HyNet easily locates the points at which the measured pressure is significantly lower than the simulated pressure, facilitating a localization of the leakage points. To maintain the optimal water pressure in the water pipeline network, it can also support the operation of valves and pumps.

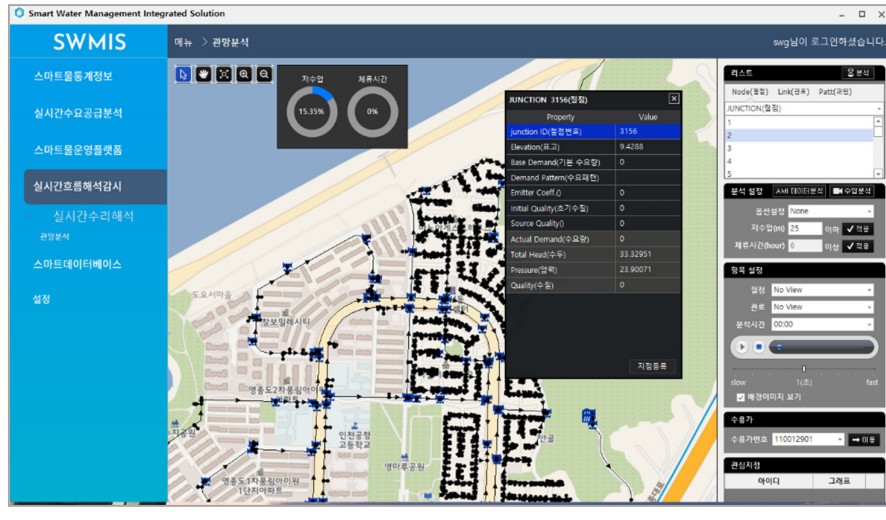

(**a**)

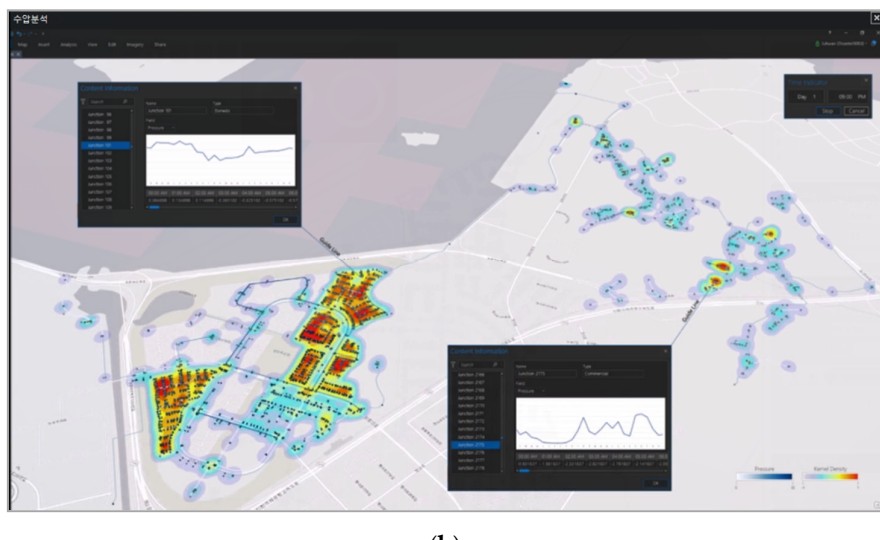

(**b**)

**Figure 15.** Smart water management integrated solution dashboard: SWG-HyNet. (**a**) Real-time pipeline network hydraulic analysis based on GIS. (**b**) Results of hydraulic analysis (nodes of interest) by hour: water pressure.

### 4.5. SWG-DBM

Smart DB management promotes the efficiency of SWG operation and management by creating daily, weekly, monthly, and yearly metadata of the input/output data of the integrated operation DB. In particular, using remote inspection water consumption data, there are functions for analyzing the water consumption patterns for each use and sending bills to consumers (Figure 16).

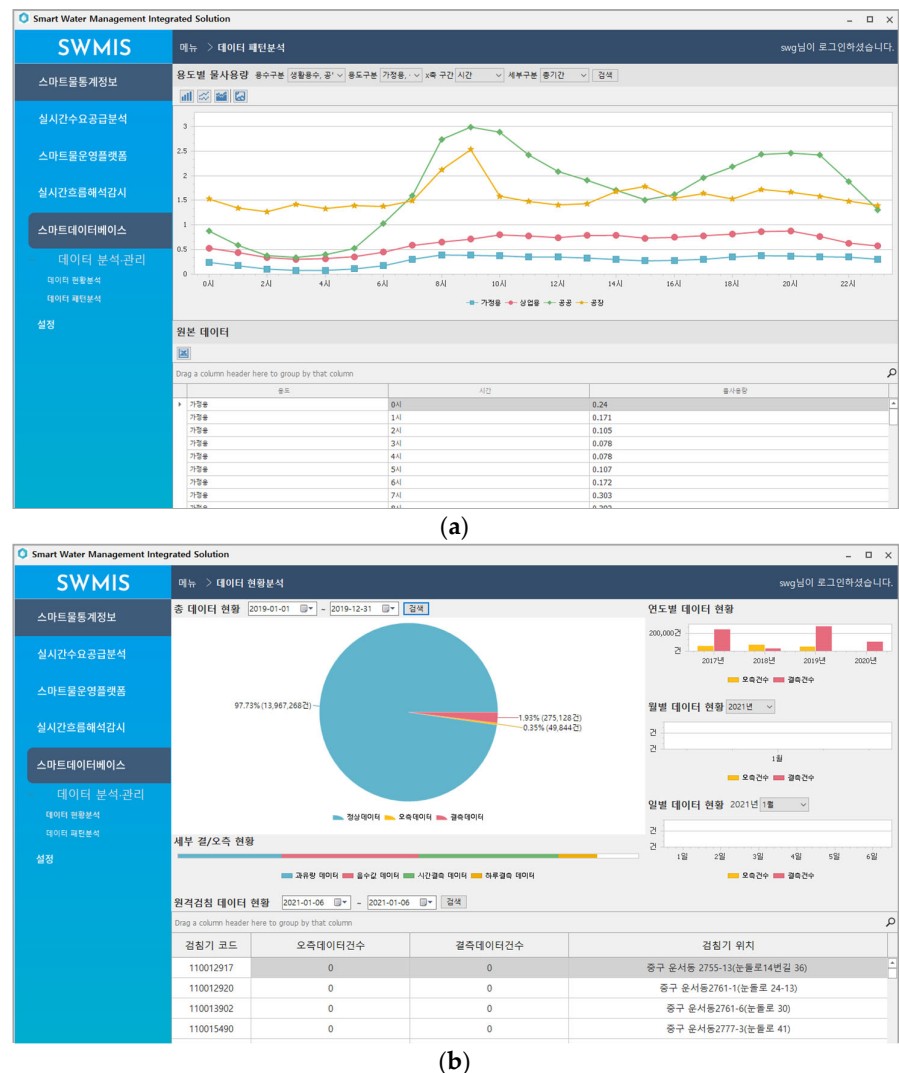

**Figure 16.** Smart water management integrated solution dashboard: SWG-DBM. (**a**) Water consumption pattern data by usage (domestic, commercial, public, and industrial). (**b**) Remote meter reading (normal, outlier, and missing values)/billing (daily and monthly).

### 4.6. SWG-App

The water information mobile app has the following functions developed: retrieval and viewing of consumer water consumption information in real time or within a specific period (day, week, month, or year), information on progressive rate of water utility and real-time water rate, welfare services for the socially vulnerable such as the elderly living alone or those without family or friends, and a community function for bi-directional communication and sharing of water information between water utility companies and consumers (Figure 17). Water utility companies can collect consumer opinions and quickly respond to incidents such as pipeline damage or freezing/bursting.

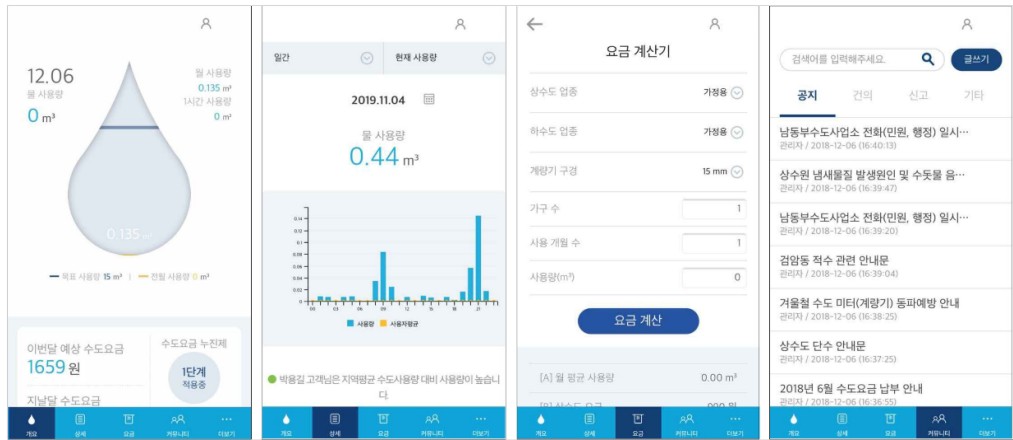

**Figure 17.** Smart water management integrated solution mobile App. dashboard: SWG-App. (main screen, real-time water consumption, charge calculation, and information reference) [21].

## 5. Discussions and Perspectives

Research has recently been on-going for the implementation of a hyper-connected smart city, which provides a solution to overall urban problems through data-based analysis and simulation by connecting all data related to the city through the Internet of Things. Based on the SWG element technologies developed in Phase 1, the SWGRG has developed a solution for integrated management of water resources in a smart city through the field operation of the SWG living lab installed at Block 112 of YeongJong Island, Incheon from 2017 to 2019. As a result, the SWGRG has outlined the following three points (data quality, unmeasured water consumption of consumers, and leak point detection) for a discussion presented on the development of an SWG-integrated operation solution through the field operation of the living lab.

### 5.1. Real-Time AMI Data Quality

The most important factor in the field operation of the SWG living lab is the quality of real-time water consumption data received through the AMI network. Poor quality data lead to unreliable analysis results. Therefore, a statistical analysis was conducted to evaluate the quality of the AMI network using the accumulated water consumption data collected hourly from November 2017 to April 2019. As shown in Figure 18, the average successful and failed receiving rates were 95.76% and 4.24%, respectively. In addition, the average failed receiving rate owing to a malfunction of the SWM was 2.78%, and the communication failure rate was 1.46%. In April 2018, a receiving rate of 82.51%, which is lower than the average, was obtained, which was due to non-payment of communication fees.

The maximum industry average of a smart meter failure rate is 5% [22], and the general smart meter failure rate is 0.5% [23]. The average failed receiving rate owing to a malfunction of the ultrasonic wave type SWM used in the field operation is lower than the maximum industry average, but higher than the rate of general smart meters. In particular, in April 2019, the failure rate reached approximately 6.24%, strongly indicating a need for improvement. In addition, the failed receiving rate owing to an SWM malfunction or failure was approximately 1.3% higher than that of communication error. In particular, it is greater than the average failed receiving rate at 18:00–21:00 h, and thus further analysis on the cause of such high failure rates is necessary.

In addition, in the event of an SWM failure, water consumption data cannot be sent even if the consumer uses water. Therefore, cases of missing data inevitably occur from the time of failure to the time of replacement. Because the collected data are the cumulative water consumption data and are counted from zero, a correction is required

to calculate the water consumption before and after SWM replacement. Water consumption can be estimated through a linear regression analysis of the mass curve from the time of failure to the time of replacement; however, because it shows an inaccurate rate of consumption, proper measures for improvement are required.

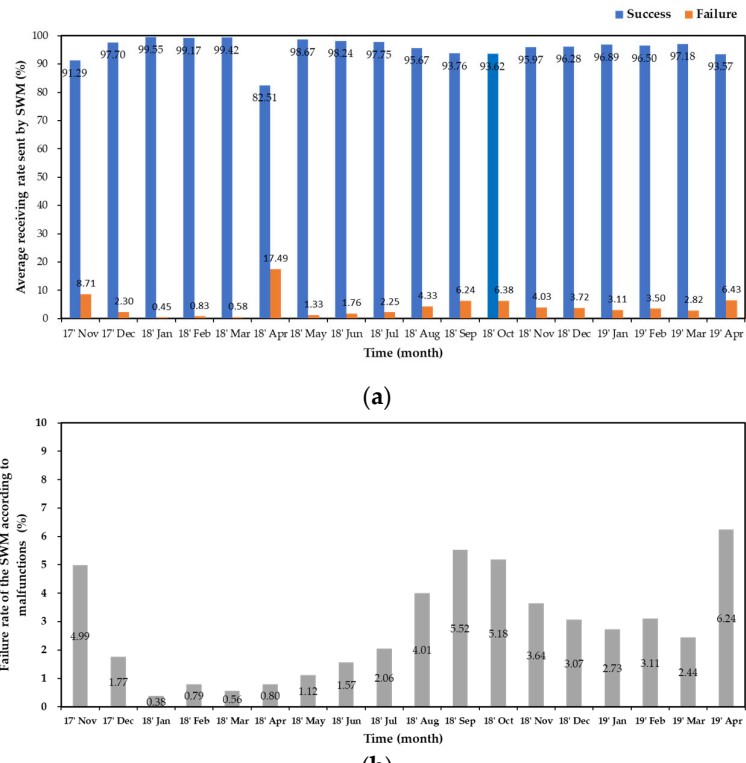

**Figure 18.** Results of real-time AMI data monitoring. (**a**) Average receiving and failure rates. (**b**) Meter failure rate.

*5.2. Unmeasured Water Consumption of Customers*

The installation of an SWM can encourage savings in water consumption from consumers, and the necessity for an SWM installation to achieve proper water management in practice is increasing for the socially vulnerable; however, consumer acceptance is required. There are currently 958 end-users on Block 112 of YeongJong Island, but SWMs are installed and operated in only 527 locations. Approximately 45% of the total amount of water use is measured by mechanical meters. The water withdrawal by water source, the amount of water filtration, and the optimal water level of the distributing reservoir are determined according to the forecast demand from the end-users. Therefore, more accurate demand forecasting is possible only when data are acquired in real time at all 958 end-user locations. In the field operation of the living lab, the real-time demand of unmeasured customers could not be obtained, and thus a demand forecast was conducted using water consumption data, monthly bill using mechanical meters, pipe diameter, usage, and water demand patterns according to the day of the week. However, it is difficult to expect high reliability for 45% of the water consumption data estimated using 55% of the real-time data. Therefore, there is a pressing need to encourage the installation of SWMs.

*5.3. Leak Point Detection*

Locating the pipeline point where the leak occurs and carrying out prompt repair work is an important task to reduce non-revenue water. Because the pipelines are buried

in the ground and are difficult to access, abnormalities are detected by ultrasonic waves or vibrations, or a water pipeline network analysis is used. Such an analysis is largely divided into pressure- and demand-driven approaches [24]. A pressure-driven water pipeline network analysis has recently drawn attention because water demand can be modeled as a function of the pressure conditions. However, it is considerably difficult in both models to predict pipeline abnormalities owing to various uncertainties. In general, a demand-driven water pipeline network analysis has been used to calculate the appropriate pressure by applying the maximum daily water consumption per person according to the population for the design of a new city. With more precise demand forecasts using an SWM and the development of water-pressure sensors, pipeline leak detection research has recently been conducted through a demand-driven water pipeline network analysis. In a living lab field operation, water pressure was measured by installing a portable water pressure meter at the fire hydrant in block 112 four times. In addition, as a result of comparing the water pressure calculated through the water pipeline network analysis using the real water demand and the water pressure measured by a portable water pressure meter, a meaningful result was obtained indicating that the water pressure simulation was possible only with the demand data (Figure 19). However, to detect the leakage point through a water pipeline network analysis, research is needed with the installation of more water pressure meters.

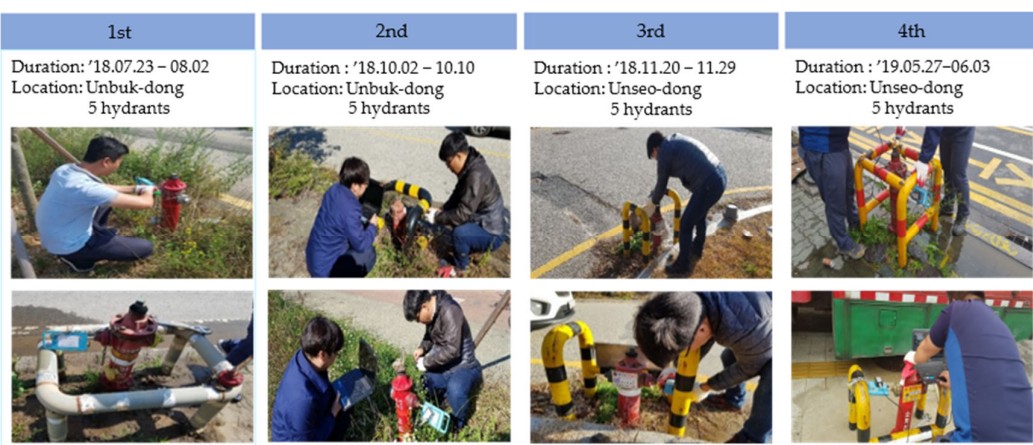

**Figure 19.** Measurement of actual water pressure in fire hydrants using a portable water pressure meter.

## 6. Conclusions

It is increasingly difficult to supply safe and sustainable water to consumers by current WDNs due to insufficient supply, deteriorating water quality, increased energy consumption, and climate change. To solve these problems, SWG integrates ICT technology and conventional water management system. Supply efficiency can be improved by solving the uncertainty of the existing manpower meter reading and alleviating water problems.

Therefore, research on the application of SWG is being conducted all over the world. In Korea, SWGRG was launched in 2012 to build an intelligent water management system using alternative water resources with the aim of reducing water and energy and built the living lab for the SWG demonstration operation. Living Lab was built in block 112 of YeongJong Island, Incheon, where Incheon International Airport, the hub of Northeast Asia, is located. Here, water is supplied through a single submarine pipeline, making it the best place to respond to water crises and build water supply systems in emergencies. SWG core technologies were developed in the living lab by combining ICT technology and water resource management technology, and demonstration operation and verification were performed.

Therefore, in this study, the core element technologies (Intelligent water source management and distribution system, Smart water distribution net-work planning/control/operation strategy establishment, AMI network and device development, and Integrated management of bi-directional smart water information) developed through the living lab demonstration operation and the development of the demonstration operation solutions (Smart water statistics information, Real-time demand-supply analysis, Decision support system, Real-time hydraulic pipeline network analysis, Smart DB management, and Water information mobile application) were introduced.

On the other hand, through the living lab demonstration operation, we found that there were several issues to be dealt with. First, one of the most important things for SWG operation is the quality of data received from AMIs. In-depth analysis of the reception failure rate due to malfunction or failure is required, and must be lowered. Second, not all consumers want to install a smart meter, so for more precise operation of the water supply system, it is necessary to forecast the water consumption of unmeasured consumers. Finally, it is necessary to examine the applicability of demand-driven pipe network analysis to the point where leakage occurs.

**Author Contributions:** K.-M.K.: Conceptualization, Investigation, Data Collection, Writing and Analysis—original draft, K.-H.H.: Managing fund, K.-S.J., G.L., and K.-T.Y.: Reviewing and Editing. All authors have read and agreed to the published version of the manuscript.

**Funding:** This Research has been performed as Project of No A-C-002 supported by K-water and the Basic Science Research Program of NRF-2020R1A2C1005554 supported by the National Research Foundation of Korea.

**Institutional Review Board Statement:** Not applicable.

**Informed Consent Statement:** Not applicable.

**Data Availability Statement:** The study did not report any data.

**Conflicts of Interest:** The authors declare no conflict of interest.

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
