# Peer review of "Smart Water Grid Research Group Project: An Introduction to the Smart Water Grid Living-Lab Demonstrative Operation in YeongJong Island, Korea"

_sustainability, doi:10.3390/su13095325_

Round 1
Reviewer 1 Report
Dear Authors,
The paper is technically sound, and certainly of interest for Sustainability readers. In the following, some comments are listed.
General comments:
- Abstract: should be reformulated in order to provide to readers an overview of the topic, of your approach and of main results
- Introduction: I suggest to better clarify goal of this paper. Furthermore, I found the sentence about “living lab” at the end of the paragraph not linked to the text.
- Conclusions: to my opinion, a dedicated section could help readers to better get quality and merits of your work
Specific comments:
- I suggest to add a table for acronyms, and to use capital letters when refer to words (e.g.: “advanced metering infrastructure” to “Advanced Metering Infrastructure”)
Author Response
Thank you very much for the invaluable suggestions.
Point 1-Abstract: should be reformulated in order to provide to readers an overview of the topic, of your approach and of main results
Response 1: We modified previous one as below.
In South Korea, in line with the increasing need for a reliable water supply following the continuous increase in water demand, the Smart Water Grid Research Group (SWGRG) was officially launched in 2012. With the vision of providing water welfare at a national level, SWGRG incorporated Information and Communications Technology in its water resource management, and built a living lab for the demonstrative operation of the Smart Water Grid (SWG). The living lab was built in Block 112 of YeongJong Island, Incheon, South Korea (area of 17.4 km2, population of 8,000), where Incheon International Airport, a hub of Northeast Asia, is located. In this location, water is supplied through a single submarine pipeline, making the place optimal for responses to water crises and the construction of a water supply system during emergencies. From 2017 to 2019, ultrasonic wave type smart water meters and IEEE 802.15.4g Advanced Metering Infrastructure (AMI) networks were installed at 527 sites of 958 consumer areas in the living lab. Therefore, this study introduces the development of SWG core element technologies (Intelligent water source management and distribution system, Smart water distribution network planning/control/operation strategy establishment, AMI network and device development, Integrated management of bi-directional smart water information), and operation solutions (Smart water statistics information, Real-time demand-supply analysis, Decision support system, Re-al-time hydraulic pipeline network analysis, Smart DB management, and Water information mobile application) were developed through a field operation and testing in the living lab.
Point 2-Introduction: I suggest to better clarify goal of this paper. Furthermore, I found the sentence about “living lab” at the end of the paragraph not linked to the text.
Response 2: We deleted the "living lab" sentence that isn’t linked to the text and modified sentences.
As above, SWG is being actively conducted worldwide. However, there are few cases of developing detailed element technologies and operation solutions through the operation of living labs worldwide. Therefore, this study introduces the development of detailed element technologies for SWG management and an operation solution for South Korea through a field operation and tests in a living lab. Also, we would like to show examples of what issues are there in living lab operation.
Point 3-Conclusions: to my opinion, a dedicated section could help readers to better get quality and merits of your work
Response 3: We added “6. Conclusions” chapter as below.
Current WDNs are increasingly difficult to supply safe and sustainable water to consumers due to insufficient supply, deteriorating water quality, increased energy con-sumption, and climate change. To solve these problems, SWG, which integrates ICT tech-nology and conventional water management system. Supply efficiency can be improved by solving the uncertainty of the existing manpower meter reading and alleviating water problems.
Therefore, research on the application of SWG is being conducted all over the world. In Korea, SWGRG was launched in 2012 to build an intelligent water management system using alternative water resources with the aim of reducing water and energy and built the living lab for the SWG demonstration operation. Living Lab was built in block 112 of YeongJong Island, Incheon, where Incheon International Airport, the hub of Northeast Asia, is located. Here, water is supplied through a single submarine pipeline, making it the best place to respond to water crises and build water supply systems in emergencies. SWG core technologies were developed in the living lab by combining ICT technology and water resource management technology, and demonstration operation and verification were performed.
Therefore, in this study, the core element technologies (Intelligent water source management and distribution system, Smart water distribution net-work planning/control/operation strategy establishment, AMI network and device development, In-tegrated management of bi-directional smart water information) developed through the living lab demonstration operation and the development of the demonstration operation solutions (Smart water statistics information, Real-time demand-supply analysis, Deci-sion support system, Real-time hydraulic pipeline network analysis, Smart DB management, and Water information mobile application) were introduced.
On the one hand, through the living lab demonstration operation, we found that there were several issues. First, one of the most important things for SWG operation is the quality of data received from AMIs. In-depth analysis of the reception failure rate due to malfunction or failure is required, and it must be lowered. Second, not all consumers want to install a smart meter, so for more precise operation of the water supply system, it is necessary to forecast the water consumption of unmeasured consumers. Finally, it is necessary to examine the applicability of demand-driven pipe network analysis to the point where leakage occurs.
Point 4-Specific comment: I suggest to add a table for acronyms, and to use capital letters when refer to words (e.g.: “advanced metering infrastructure” to “Advanced Metering Infrastructure”)
Response 4: All have been corrected to acronyms using capital letters suggested by reviewer for Point 4-specific comment.

Reviewer 2 Report
Thank you for the opportunity to review this nicely presented piece of interesting and timely research colleagues from South Korea with the title: Smart Water Grid Research Group Project: An Introduction to the Smart Water Grid Living-Lab Demonstrative Operation in YeongJong Island, Korea
* Paper is interesting for readers and it has technical sound. Today not only in South Korea is necessary in line with the increasing need for a reliable water supply following the continuous increase in water demand to develop and construct a highly efficient next-generation water management infrastructure system and verify it through a field operation in real circumstances.
* Materials & methods: The logic of the research seems sound overall. The Smart Water Grid Research Group with very well funding research incorporated information and communications technology in its water resource management, aiming at the development of core technologies for a Smart Water Grid consisting of intelligent microgrids with a demonstration and tests of the developed technologies through a field operation in a living lab. The suitability and technical standards of the used methods is applicable everywhere. Sufficient details of the methods is provided so that another researcher is able to reproduce the experiments described. But I am afraid that not all cities/coutries have the similar necessary input tools (finance) available.
* Results & discussion: The data are very well controlled and robust – 56 months, resp. 33 months of measurements, advanced metering infrastructure networks installed at 527 sites of 958 consumer areas. Authors used smart water statistics information, real-time demand-supply analysis, decision support system, real-time hydraulic pipeline network analysis, smart DB management, and water information mobile application, GIS, UI, ... all system was developed through a field operation and testing. Authors also provided relevant and current references. Discussion and conclusions are based on actual facts and figures.
* Conclusion: Authors provided adequate proof for their claims. Also they wrote about negatives and some malfunction of their research and necessary improvements for the future – smart cities water operation.
* All the references cited are relevant and adequate.
However the manuscript, in its present form needs only small corrections/ minor issues. I propose the following points:
- Abstract: there are some incorrect dividing of words (man-agement; op-eration)
- Introduction: I am not sure, that all mentioned countries - especially in the frame of European Union - have taken the lead implement a smart water management by incorporating SWG Technologies. I live in one of these countries and we are very far from this idea. And at the description of Milano-Timisoara test is not finished sentence ...AMI network was conducted
- Figure 4 - different high of some letters
- Figure 8 – better is to put the all titles in the Legend (NC, OHD, SMD) are not very known abbreviations
- Fig. 12-17 – used not english but origin language. It will be nice to put more detailed description what we can see there.
Author Response
Thank you very much for the invaluable suggestions and questions.
Point 1-Abstract: there are some incorrect dividing of words (man-agement; op-eration)
Response 1: Actually, "-" is not inserted in the middle of the words by us, but please understand that the journal form cannot be modified due to the spacing problem (insert "-") of the Office Word default setting.
Point 2-Introduction: I am not sure, that all mentioned countries - especially in the frame of European Union - have taken the lead implement a smart water management by incorporating SWG Technologies. I live in one of these countries and we are very far from this idea. And at the description of Milano-Timisoara test is not finished sentence ...AMI network was conducted
Response 2: We, SWGRG signed an MOU with the University of Perugia, Italy, and confirmed that the research on the leakage detection of water distribution network using smart sensor monitoring technique is proceeding considerably by Professor Bruno Brunone.
Reference is: https://iwaponline.com/jh/article/17/3/377/3665/Anomaly-pre-localization-in-distribution
And we also corrected to the incomplete sentences.
Point 3-Figure 4: different high of some letters
Response 3: We corrected to the different high for Figure 4
Point 4-Figure 8: better is to put the all titles in the Legend (NC, OHD, SMD) are not very known abbreviations
Response 4: We put the all titles in the legend instead of acronyms as suggested by Reviewer.
Point 5- Fig. 12-17: used not English but origin language. It will be nice to put more detailed description what we can see there.
Response 5: We put more detailed description to the caption from Fig 12. to Fig. 17.
